# Unraveling the Relative Contributions of Deterministic and Stochastic Processes in Shaping Species Community Assembly in a Floodplain and Shallow Hillslope System

**Gustavo Enrique Mendoza-Arroyo** [1,*][iD]**, René Efraín Canché-Solís** [2,3]**, Alejandro Morón-Ríos** [4][iD]**, Mario González-Espinosa** [5][iD] **and Moisés Méndez-Toribio** [6,7][iD]

1    Departamento de Ciencias Básicas, Tecnológico Nacional de México, Instituto Tecnológico de Chiná, Chiná 24520, Campeche, Mexico
2    Instituto Everest de Sinaloa, Mazatlán 82129, Sinaloa, Mexico; 2104013@educacioneverest.com or rene.cs@campeche.tecnm.mx
3    Departamento de Ingeniería Química y Bioquímica, Tecnológico Nacional de México, Instituto Tecnológico de Campeche, Lerma 24500, Campeche, Mexico
4    Departamento de Conservación de la Biodiversidad, El Colegio de la Frontera Sur, Lerma 24500, Campeche, Mexico; amoron@ecosur.mx
5    Departamento de Conservación de la Biodiversidad, El Colegio de la Frontera Sur, San Cristobal de las Casas 29290, Chiapas, Mexico; mgonzale@ecosur.mx
6    Red de Diversidad Biológica del Occidente Mexicano, Centro Regional del Bajío, Instituto de Ecología A.C., Pátzcuaro 61600, Michoacán, Mexico; moises.mendez@inecol.mx
7    Consejo Nacional de Humanidades, Ciencias y Tecnologías, Benito Juárez 03940, Ciudad de Mexico, Mexico
*    Correspondence: gustavo.ma@china.tecnm.mx

**Abstract:** Understanding the process underlying species coexistence is crucial in ecology. This challenge is relevant in tree communities inhabiting contrasting abiotic conditions, such as lowland floodplain and shallow hillslope karstic systems. We examined the influence of topographic variables and spatial factors on the structure of tree communities in the karstic system in Calakmul, Mexico. We measured 7050 trees (diameter at breast height $\geq$ 3 cm) in 152 circular plots and generated seven topographic variables from a digital elevation model. We employed redundancy analysis and variance partitioning to test the effects of environmental and spatial factors on tree communities. In addition, we used the null Raup–Crick model to uncover the relative importance of the deterministic and stochastic processes driving community assembly. Our study revealed significant floristic distinction between seasonally flooded and upland forests. The topographic wetness index (TWI) contribution to explaining the floristic differentiation in the studied tree assemblages was greater than that of the other topography-related variables. The explanatory power of the environmental and spatial factors varied slightly between datasets. The null model indicated a predominant influence of deterministic over stochastic processes. Our findings reaffirm the role of seasonal flooding as an abiotic filter. Additionally, the TWI can serve to identify flood-prone conditions within shallow depressions. The preservation of adjacent seasonally flooded and upland forests is relevant for the maintenance of tree diversity in the karst of the Yucatan Peninsula, since flooding drives the distribution of species.

**Keywords:** dispersal; environmental heterogeneity; karst; lowland forest; Raup–Crick

## 1. Introduction

A central goal in ecology is to understand the processes that underlie species community assembly [1]. Stochastic (such as dispersal limitation) and deterministic (such as habitat filtering) processes are responsible for maintaining species coexistence and diversity in tropical forests [2]. The neutral theory [3] is a hypothesis positing that, at a regional scale, dispersal limitation is a key process shaping plant communities [4,5]; namely, the dispersion between adjacent sites generates spatial autocorrelation [3]. Although species

distribution should not be related to the environment [4], it has been suggested that, if dispersal limitation is a determinant of species community assembly, similarity in floristic composition should decrease with an increase in geographic distance between pairs of plots [3,6].

In contrast, deterministic processes, such as habitat filtering [7,8], influence species sorting between habitats [9]. This perspective predicts that community similarity declines with increasing environmental heterogeneity. Therefore, the coexistence is possible because trees show habitat-specific adaptations and habitats contain species that differ in their niche requirements [2,10]. Coexistence is a consequence of differences in resource use among interacting species [11]. Niche partitioning among species or high environmental heterogeneity can enable species with similar niches to be restricted to somewhat different environments, allowing for their coexistence [12].

In the context of lowland floodable areas and upland forests, however, the extent to which these processes contribute to species community assembly remains inconclusively determined [13–19]. Waterlogging stress creates a pattern of disturbance between inundated and non-inundated areas [20]. In lowland areas, edaphic- and topographic-related variables drive seasonal flooding, generating anoxic stressful environments where species require special adaptations to survive the resulting conditions [21,22]. This habitat specialization leads to floristic differentiation between flooded and non-flooded forests [20,23–26] and contributes to maintaining a high regional diversity of plant species [27].

In Amazonian flooding forests, environmental factors such as substrate properties, topographic-related variables, hydroperiods, and their temporal dynamics have been suggested to drive the variation in floristic composition [28,29]. Edaphic characteristics [30] and elevation, convexity, and slope steepness also play a crucial role in defining the distribution of trees in non-flooded environments [31]. Related topographic variables like the topographic wetness index (TWI) [32] are key determinants of the distribution of plants in flooding environments [33]. Under local restrictive conditions [34] and disturbed conditions, environmental filters are expected to operate [35]. Therefore, seasonal flooding can act as an environmental filter [22,36]. In this case, seasonally flooded forests are naturally disturbed environments [19] that can be used as study models to discern the relative importance of the processes involved in the structuring of communities [9].

The southern portion of the Yucatan Peninsula (YP) is a karstic landscape, a mosaic formed by plains and hills whose origin is limestone or carbonate rock [37], useful as a model for investigating the relative importance of ecological processes maintaining species diversity. This is due to the presence of seasonally flooded depressions interspersed with low elevation hills [38,39]. The elevation gradients in the Yucatan Peninsula between stands of the seasonally flooded forest and the upland forest are typically 10–20 m [40,41], and usually within a distance of less than 500 m. Despite the short geographic distances involved, these forests can be clearly differentiated based on their topographical, geomorphological, and edaphic factors [42–44]. Seasonally flooded lowland flats can experience environmental stress that lasts for 1–6 months during the rainy season [45,46]. Furthermore, the distribution of seasonally flooded stands and upland forest fragments is interspersed, which could potentially limit seed dispersal across habitats where these two forest types occur (i.e., depressions and upland areas).

In the context of a seasonally flooded karstic system, this study aims to examine the underlying processes that structure two contiguous tree communities inhabiting contrasting abiotic conditions in floodplains and shallow karstic hillslopes. We hypothesize firstly that the environmental conditions contributing to seasonal flooding in karstic depressions reflect the operation of an environmental filter, which can lead to conspicuous differences in floristic composition between the two forest types. Secondly, we similarly expect that topographic-related variables to have a greater contribution than spatial factors in explaining the distribution of tree species, and finally, through the application of a null model, we aim to reveal the relative contribution of environmental filtering and dispersal limitation on the structure of the tree community.

## 2. Materials and Methods

### 2.1. Study Area and Forest Types

The study was carried out in the southern region of the Yucatan Peninsula (YP), Mexico, which lies between the coordinates of 18°31'–19°03' N and 89°48'–90°10' W. Specifically, the study area is located within the municipality of Calakmul in the state of Campeche. The landscape is predominantly karstic and spans approximately 2340 km² (39 × 60 km), which includes parts of the Balam-kin and Balam-kú natural protected areas (Figure 1). The climate in the study area is warm sub-humid with a monsoonal rainfall pattern [46,47]. The environmental variation for the last 65 years, recorded at the meteorological stations near the sampling sites, indicates that the average annual temperature ranges between 26.3 and 26.7 °C and the total annual precipitation ranges from 908.6 to 1396 mm (https://smn.conagua.gob.mx/es/climatologia, accessed on 7 September 2023).

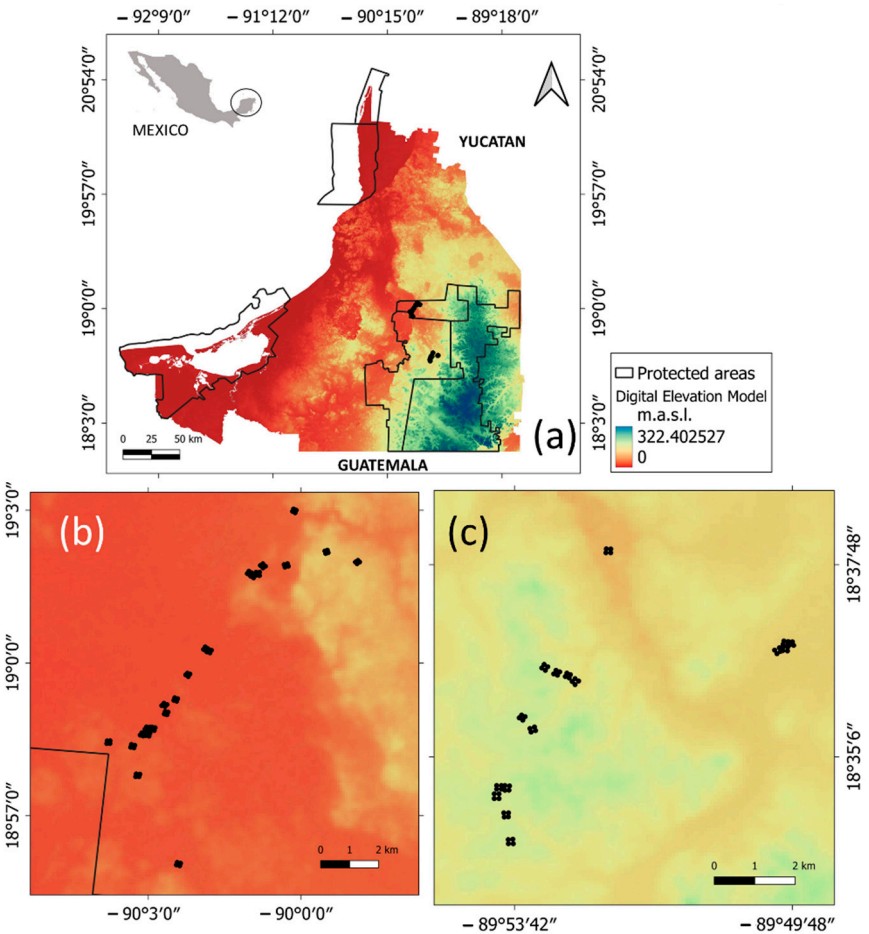

**Figure 1.** Study area and location of sampling vegetation plots (black dots) in the municipality of Calakmul, state of Campeche, Mexico. (**a**) State of Campeche; (**b**) Balam-kin protected area; (**c**) Balam-kú protected area.

The upland forest in the study area is located in low hills with rocky, well-drained leptosols and is primarily dominated by *Guaiacum sanctum* L. (guayacan) [45,48]. The central part of the state of Campeche in Mexico is home to the largest remaining populations of *G. sanctum* [49,50]. In the YP, the presence of *G. sanctum*, along with *Thouinia paucidentata* Radlk. and *Manilkara zapota* (L.) P. Royen, is associated with old-growth forests in good conservation conditions [51]. The seasonally flooded forest in the YP develops in depressions called "bajos" [40,44,52]. These lowland flats are characterized by gleysols that remain flooded for long periods due to their high clay content [38,53]. However, detailed data on the physical and chemical soil variables from lowland sites are scarce, which results

in difficult comparisons with nearby upland sites (Table S1) [38,53,54]. The seasonally flooded forest is a unique formation endemic to the YP [55,56]. The dominant tree species in the forest is *Haematoxylum campechianum* L., locally known as "palo de tinte", which forms mono-dominant stands referred to as "tintales" [57].

### 2.2. Field Sampling and Data Collection

A comprehensive tree inventory was conducted in the study area using 152 circular plots, of which 60 were in flooded forest stands and 92 in upland forest. All trees with a diameter at breast height (DBH) of 10 cm or more were counted and measured within a 500 m$^2$ plot (diameter of 25.23 m), while individuals with a DBH ranging between 3.0 and 9.9 cm were counted and measured within circular subplots of 100 m$^2$ (diameter of 11.28 m) inside the larger plots. Tree height was measured with a Haglöf® digital clinometer type HCH, which provided a precision of 0.1 degree [58]. DBH was measured using a Haglöf® diametric tape.

### 2.3. Topographic Variables

We calculated seven topography-related variables for each of our study plots (Table S2), including aspect, elevation, roughness, slope, topographic position index (TPI), topographic roughness index (TRI), and topographic wetness index (TWI) from a digital elevation model (DEM) with a resolution of 15 m obtained from INEGI (accessed on 7 September 2023). The DEM is available for download free (https://www.inegi.org.mx/app/geo2/elevacionesmex, accessed on 22 August 2023), with no registration required. Aspect was determined as the direction of the steepest slope within each quadrat. Elevation (a.s.l.) was extracted from DEM. Roughness was calculated as the maximum difference in elevation between a focal cell and its eight adjacent cells [59]. The topographic position index (TPI) quantifies the difference between the elevation value of a given cell and the average elevation of neighboring cells within a defined radius [60]. This information is useful for identifying low-lying areas and depressions that are more likely to be wet [61]. A positive TPI value indicates a higher elevation (ridge) than the surrounding area, while a negative value indicates a lower elevation (valley), which is typically indicative of wetland areas [62].

The terrain ruggedness index (TRI) calculates the mean of the absolute differences in elevation between a focal cell and its eight surrounding cells, quantifying the total elevation change across a 3 × 3 cell area. Flat areas with little variation have a TRI of zero, while mountainous regions with step ridges have positive values [63]. The topographic wetness index (TWI) [32] is a commonly used proxy for soil moisture content, indicating the potential soil water storage condition of a pixel. The TWI combines the local upslope contributing area and slope to quantify the topographic control on hydrological processes [64]. The index is derived from three key components: total catchment area, flow width, and slope gradient [65]. The TWI predicts water accumulation by describing the shape of the land at any given point in the landscape as the ratio of the uphill area from which water would flow into that point to the local slope at that point [66]. As such, the TWI describes the spatial distribution saturation zones, making it a common surrogate for soil water content [67]. High TWI values are typical of converging, flat terrains and indicate a high potential for saturation, while low values are typical of steep and diverging areas and indicate a low potential for soil saturation. We calculated slope, roughness, and aspect using the corresponding tools in QGIS 3.26.3. To derive the TWI and other derivative variables such as the topographic position index (TPI) and terrain ruggedness index (TRI), we used the morphometry and hydrology modules. All the modules used are open source and are available for free in the System for Automated Geoscientific Analyses (SAGA GIS, v. 2.3.2 https://saga-gis.sourceforge.io/en/index.html, accessed on 22 August 2023).

*2.4. Data Analysis*

A non-metric multidimensional scaling (NMDS), based on the Bray–Curtis index as a dissimilarity distance using abundance data, was performed to distinguish differences between the stands of seasonally flooded forest and upland forest. This ordination method was realized using the vegan package [68]. Species that contributed the most to community dissimilarity among stands were identified using SIMPER (similarity percentage) analysis [69]. To examine whether the floristic composition significantly differed among the forests (upland vs. seasonally flooded), we employed one-way analysis of similarity (ANOSIM) with 999 permutations [69]; both analyses were performed in R using the vegan package [68] in R software (version 3.6.2, R Core Team, http://www.R-project.org, accessed on 3 September 2023).

To determine the contribution of environmental variables and spatial factors to species composition, we conducted a redundancy analysis (RDA) [70]. We utilized Hellinger-transformed abundance data (relative abundances) as the response matrix [71]. For the spatial explanatory matrix, we calculated the spatial distance among all plots using latitude–longitude data and transformed it to rectangular principal coordinates of neighborhood matrices (PCNM) using the pcnm function of the vegan [72,73] in R software (version 3.6.2, R Core Team, http://www.R-project.org, accessed on 3 September 2023). The explanatory matrix included spatial distance and seven environmental variables: aspect, elevation, roughness, slope, topographic position index (TPI), topographic roughness index (TRI), and topographic wetness index (TWI).

We conducted variance partition analysis [74] to determine the effects of dispersal limitation, represented by pure spatial component, and abiotic filtering, represented by the sum of purely abiotic component and spatially structured abiotic component. We employed a forward model selection with the *ordiR2step* function (200 permutations) and variance inflation factor (VIF < 10) with the *vif* function to select only significant spatial and environmental explanatory variables for the final model [75]. This procedure was carried out to choose a parsimonious RDA model with the highest adjusted $R^2$. The varpart function was employed to determine the extent of the pure environmental (explained by abiotic factors only), pure spatial (explained by spatial factors), spatial component of environmental influence, and undetermined variables (residual). To test the statistical significance of each fraction, we used the anova.cca function available in the vegan package [68] in R software (version 3.6.2, R Core Team, http://www.R-project.org, accessed on 3 September 2023).

To disentangle the importance of deterministic processes from stochastic processes underlying assembly, we utilized a null model [76]. This model includes a modification of the Raup–Crick dissimilarity index [77], which is not influenced by differences in local species richness. The index uses a randomization procedure to compare the observed number of species occurring in both sites with the distribution of co-occurrences from 1000 random replicates. This approach makes it possible to infer the mechanisms that may determine community structure by comparing observed beta diversity against theoretical beta diversity in stochastically structured assemblages [76].

Values closer to +1 indicate that two communities are more dissimilar, sharing fewer species than expected by chance, which suggests the preponderance of biotic filter (competition) or spatial aggregation of the species due to dispersal limitation, i.e., very low dispersal among sites. Metric values close to -1 indicate that two communities are more similar than expected by chance, while values close to 0 indicate that dissimilarity between two communities does not differ from null expectation, suggesting that community assembly is highly stochastic, and dispersal is high among communities [76].

The value of this metric provides some indication of the possible underlying mechanisms of community assembly, particularly the degree to which deterministic processes create communities that deviate from those based on stochastic (null) expectations. We obtained the null estimates with 9999 randomizations using the raupcrick function available in the vegan package [68] in R software (version 3.6.2, R Core Team, http://www.R-project.org, accessed on 3 September 2023).

## 3. Results

### 3.1. Tree Community Composition

In total, we recorded 7050 individual trees and identified 139 tree species in the two forest types (Table 1). The NMDS ordination indicates that all flooded forest assemblages analyzed (DBH ≤ 10 cm, DBH ≥ 10 cm, and all individuals) tended to be more similar in their species composition (Figure 2). When we examined similarity in each assemblage separately, we found markedly that both forest types maintain different tree species assemblages (ANOSIM R = 0.464 and 0.534 for all individuals and DBH ≥ 10 cm, respectively). The SIMPER analysis of the all-individuals dataset showed that there was an 83.25% dissimilarity between the forest types (Table 2). The differentiation of the all-individuals dataset assemblage was determined by a group of five species: *G. sanctum* (7.08%), *Lonchocarpus yucatanensis* (6.91%), *M. zapota* (6.82%), *Coccoloba cozumelensis* (6.10%), and *H. campechianum* (5.87%), which together contributed 32.78% to the differentiation.

**Table 1.** Description of seasonally flooded forest and upland forest in the Calakmul region, Campeche, Mexico. Density estimates are based on tree stems with DBH ≤ 10 cm and DBH ≥ 10 cm.

|  | Seasonally Flooded Forest | Upland Forest |
|---|---|---|
| Number of plots | 60 | 92 |
| Number total of stems measured | 2823 | 4227 |
| Number of stems (DBH ≤ 10 cm) | 1112 | 1091 |
| Density (tree stems·ha$^{-1}$) | 1853.33 | 1185.87 |
| Number of individuals (DBH ≥ 10 cm) | 1711 | 3136 |
| Density (tree stems·ha$^{-1}$) | 570.33 | 681.74 |
| Sobs | 69 | 107 |
| Schao2 | 94 | 138 |
| Sobs/Schao2 (%) | 72.3 | 77.5 |

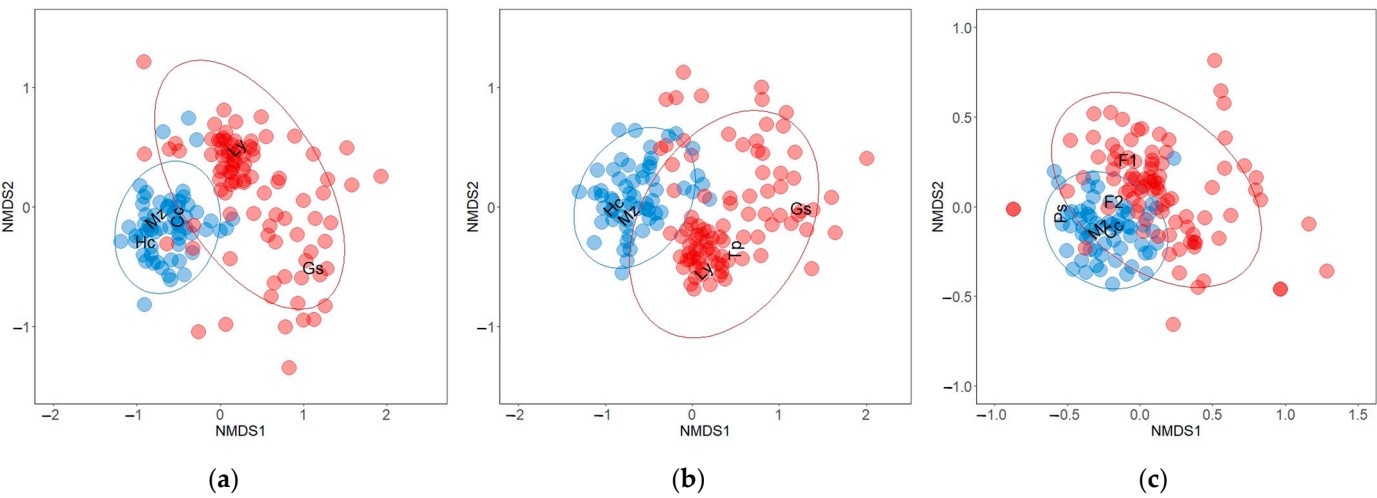

(a)                     (b)                     (c)

**Figure 2.** Non-metric multidimensional scaling (NMDS) using the Bray–Curtis dissimilarity metric to distinguish floristic composition differences between seasonally flooded forest (blue circles) and upland forest (red circles) trees within the municipality of Calakmul in the state of Campeche, Mexico. Ellipses encompassed 95% confidence interval. (**a**) All-individuals dataset (7050 tree stems), (**b**) DBH ≥10cm (4847 tree stems) and (**c**) DBH ≤10cm (2203 tree stems). Scientific name abbreviations: *Guaiacum sanctum* (Gs); *Lonchocarpus yucatanensis* (Ly); *Manilkara zapota* (Mz); *Coccoloba cozumelensis* (Cc); *Haematoxylum campechianum* (Hc); *Thouinia paucidentata* (Tp); *Psidium sartorianum* (Ps); Fabaceae 1 (F1); Fabaceae 2 (F2).

**Table 2.** Results of SIMPER (similarity percentage, dissimilarity), ANOSIM (analysis of similarities) and NMDS (non-metric multidimensional scaling) to compare floristic composition of trees recorded in a seasonally flooded forest and upland forest within the municipality of Calakmul in the state of Campeche, Mexico.

| Dataset | SIMPER (% Dissimilarity) | ANOSIM Global R | NMDS Stress Level ($R^2$ adj. Non-Linear Fit/$R^2$ adj. Linear Fit) |
|---|---|---|---|
| All individuals | 83.25% | 0.464 ($p = 0.001$) | 0.20 (0.95/0.81) |
| DBH $\geq$ 10 cm | 86.84% | 0.534 ($p = 0.001$) | 0.20 (0.95/0.82) |
| DBH $\leq$ 10 cm | 84.99% | 0.240 ($p = 0.001$) | 0.20 (0.95/0.83) |

### 3.2. Important Local Environmental Variables and Spatial Components

The redundancy analysis (RDA) indicated that the TWI was significantly related to floristic differentiation in all the types of datasets studied. Additionally, variables such as aspect, TPI and roughness showed significant relationships with both the all-individuals dataset and the DBH $\geq$ 10 cm individuals dataset. In contrast, variables such as elevation, slope, and TRI were not found to significantly structure communities. Based on the selection approach, the spatial factor analysis revealed that PCNM3, PCNM4, PCNM5, and PCNM6 were significantly related in all three assemblies analyzed. PCNM2 was significant in both the all-individuals dataset and the DBH $\leq$ 10 cm individuals dataset. The first PCNM vector (PCNM1) did not significantly structure any communities (Table 3).

**Table 3.** Results of the forward selection method in the redundancy analysis (RDA) for trees recorded in a seasonally flooded forest and upland forest within the municipality of Calakmul in the state of Campeche, Mexico. TPI: topographic position index; TRI: topographic roughness index; TWI: topographic wetness index; PCNM: principal coordinates of neighboring matrices. (-): variables that were not selected by the selection method.

| Dataset | All Individuals | | | Stems > 10 cm | | | Stems < 10 cm | | |
|---|---|---|---|---|---|---|---|---|---|
| | F | $p$-Value | $R^2$ adj. | F | $p$-Value | $R^2$ adj. | F | $p$-Value | $R^2$ adj. |
| Aspect | 3.19 | 0.008 | 0.17 | 3.11 | 0.008 | 0.15 | - | - | - |
| Elevation | - | - | - | - | - | - | - | - | - |
| Slope | - | - | - | - | - | - | - | - | - |
| Roughness | 2.13 | 0.048 | 0.20 | 2.37 | 0.034 | 0.18 | - | - | - |
| TPI | 2.41 | 0.018 | 0.18 | 2.31 | 0.03 | 0.17 | - | - | - |
| TRI | - | - | - | - | - | - | - | - | - |
| TWI | 15.27 | 0.002 | 0.09 | 15.69 | 0.002 | 0.09 | 5.94 | 0.002 | 0.08 |
| PCNM1 | - | - | - | - | - | - | - | - | - |
| PCNM2 | 2.38 | 0.024 | 0.17 | - | - | - | 4.11 | 0.02 | 0.10 |
| PCNM3 | 3.07 | 0.008 | 0.15 | 3.32 | 0.004 | 0.17 | - | - | - |
| PCNM4 | 8.15 | 0.002 | 0.13 | 8.13 | 0.002 | 0.13 | 9.29 | 0.002 | 0.05 |
| PCNM5 | 3.41 | 0.002 | 0.14 | 3.68 | 0.008 | 0.14 | 2.14 | 0.05 | 0.11 |
| PCNM6 | 2.44 | 0.02 | 0.19 | 2.32 | 0.02 | 0.19 | - | - | - |

### 3.3. Relative Role of Environmental and Spatial Factors

The variance partitioning analysis revealed that jointly the environmental variables, spatial components, and their shared effects accounted for 19.93% of the community variation in the all-individuals dataset, 19.70% in the DBH > 10 cm individuals dataset, and 12.63% in the DBH < 10 cm individuals dataset. The relative importance of environmental and spatial factors varied slightly among the datasets. The shared fractions accounted for the smallest amount of variation (2.63%–4.91%) and the residual or unexplained variance accounted for over 80.07% (Table 4).

**Table 4.** Variation partitioning, pure environmental component, shared component and pure spatial component. The entire component is based on $R^2$ adjusted values indicated as proportions of the overall variation. The residual refers to unexplained variance.

| Dataset | Environmental | Shared | Spatial | Residual |
|---|---|---|---|---|
| All individuals | 6.89% | 4.64% | 8.40% | 80.07% |
| DBH ≥ 10 cm | 6.96% | 4.91% | 7.84% | 80.30% |
| DBH ≤ 10 cm | 3.01% | 2.63% | 6.98% | 87.37% |

*3.4. Processes Underlying Assembly*

The beta Raup–Crick values obtained from the null model (Figure 3) for the all-individuals dataset ranged from 0.12 to 0.72 ($\bar{x}$ = 0.33), indicating that the communities tend to be more dissimilar than expected by chance (beta RC $\bar{x}$ = 0.36 in DBH ≥ 10 cm individuals dataset and beta RC $\bar{x}$ = 0.28 in DBH ≤ 10 cm individuals dataset).

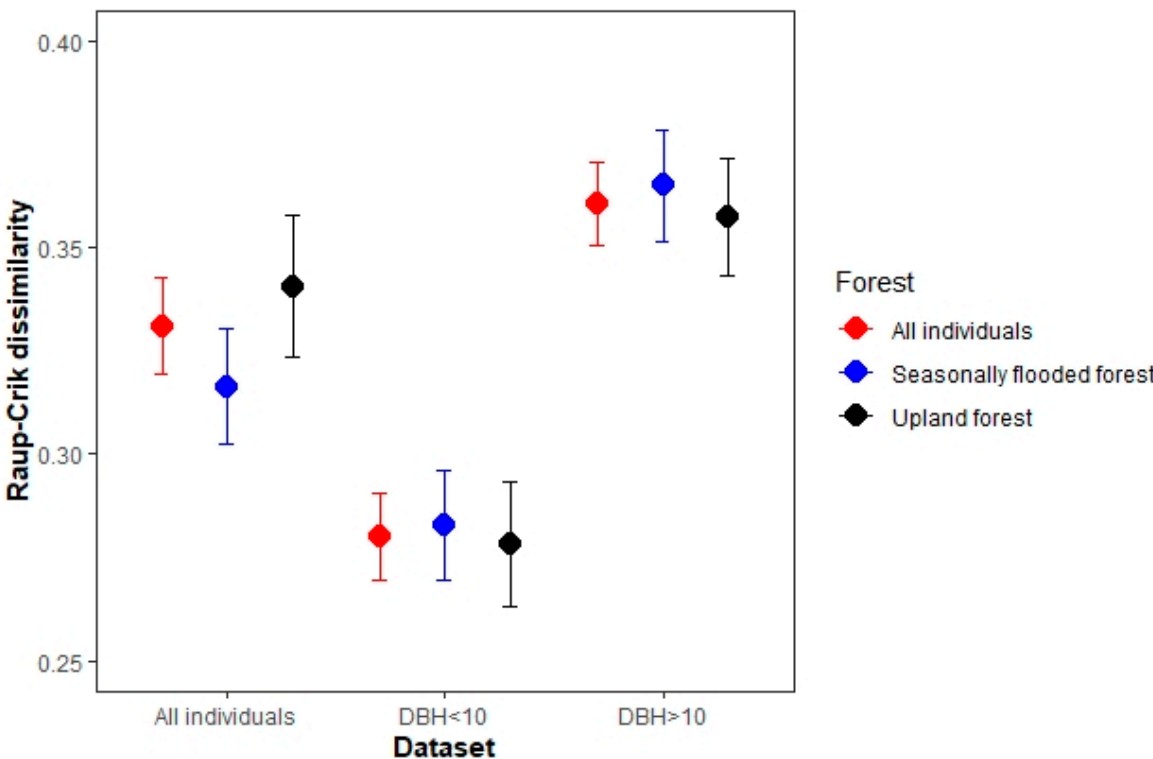

**Figure 3.** Mean dissimilarity (±S.E.) according to Raup–Crick metric among sampling plots (beta Raup–Crick, BRC) in three different assemblies within the municipality of Calakmul in the state of Campeche, Mexico.

## 4. Discussion

This study aimed to assess the influence of topography-related factors and geographical distance on the organization of two tree communities located in the southeast of the YP region. The study area comprises a floodplain and hillslope karstic system that spans lowland depressions with seasonally flooded forests to well-drained shallow hills where upland forests develop. Our results showed that the floristic distinction between the two studied forest types suggests that, regardless of the type of assemblage data, the seasonally flooded forest community may be shaped by environmental filtering.

We found a grouping pattern in both the seasonally flooded forest and the upland forest plots, indicating a high similarity in their floristic composition within each forest type. This, in turn, resulted in a marked dissimilarity and high turnover of species between the two forest types. Gley-type soils, which are prevalent in the flooded depressions

of the Calakmul region, have high clay content [38,53], favoring water retention and seasonal flooding due to local precipitation accumulation and surface runoff from rainwater, characteristic of the YP karstic landscape [39]. As such, the terrain depressions where the seasonally flooded forest develops can be considered stressful habitats due to their abiotic conditions, imposed by seasonal waterlogging [38,45,53,78]. In this sense, our results show that the floristic distinction between the seasonally flooded and upland forests in the Calakmul region is possibly due to seasonal flooding. Such a pattern of floristic differentiation is similar to that reported for the flooded and terra firme forests of Amazonia [15,79].

In order to investigate the influence of seasonal flooding on floristic differentiation between forest types, we utilized abiotic factors related to topographic features, which have been previously used as a proxy for hydrological processes. The topographic wetness index (TWI) is a reliable flood proxy [80] and has contributed significantly to revealing the processes influencing community assembly [81]. Our results showed that the TWI is a good predictor of the floristic distinction between the studied assemblages, and that seasonal flooding can generate a strong environmental filtering and lead to floristic differences in the forests studied, which is consistent with the findings of previous studies [82].

The variable selection procedure showed that elevation, slope (inclination), and the TRI did not influence tree distribution; this is possibly due to the low relief of the Calakmul region and particularly in the portions of the YP close to the Gulf of Mexico, where the relief is very shallow [39]. In our study, the highest elevation recorded was approximately around 136 m a.s.l., where the upland forest stands develop. The highest areas (~400 m a.s.l.) are located in the center of the YP [83]. These variables are more relevant in conditions of steeper altitudinal gradients, such as those found in other karstic environments [84].

In Amazonian flooded forests, floristic differentiation has been attributed to the influence of environmental filtering [15–18,28]. In our study system, we suggest that seasonal flooding significantly influences species sorting, particularly in lowland depressions, affecting the recruitment of tree species dispersing between upland and flooded areas. This is supported by the low floristic similarity observed between the tree assemblages in these two habitats (Figure 2). As we have noted previously, the forests studied can be distinguished by their edaphic and geomorphological characteristics (Tables S1 and S2); as well as for their structure and floristic composition. We found that the floristic distinction is indicated by the contribution of the typical species of each environment, mainly by *G. sanctum* (upland forest) and *H. campechianum* (seasonally flooded forest). *M. zapota* is a species of ecological importance that is notable in the forests of Calakmul [51] and occurs in both environments (seasonally flooded and upland forest). However, it may be experiencing phenotypic plasticity, since in the floodplains it is able to establish itself but with individuals that have reduced diameters (DBH $\leq$ 20 cm), in the opposite way to that in which it occurs in the hills, where individuals show larger diameters (DBH $\geq$ 35 cm).

The absence of predictive environmental variables can enhance the contribution of the spatial component. Therefore, it is possible that the contribution of the spatial component was slightly higher than the environmental component. However, the amount of variation explained by the spatial component cannot be considered as evidence of the predominance of stochastic processes, even though the forests studied are separated by distances of less than 500 m.

The unmeasured but potentially significant environmental variables may have contributed to decreasing the amount of unexplained variation. Future studies should take into consideration the physical and chemical variables of the soil to obtain a better explanation of the variance and more accurate understanding of the assembly mechanisms in such systems. However, the evidence obtained is insufficient to rule out the influence of spatial factors, such as dispersal limitation (i.e., spatial distance), which is a prominent factor in species turnover between seasonally flooded and upland forests. We did not include any seed functional traits, which are a proxy for dispersal ability [85]; future studies including

seed traits should improve our understanding about the role of dispersal limitation in this karstic system, limited by short distances.

Moreover, the null model proposes that a significant number of plots exhibited greater dissimilarities than could be expected by chance. The observed pattern of fewer shared species than could be expected by chance implies that the turnover of species between the two forest types is barely influenced by dispersal limitation. However, we cannot discount its plausible effect on species differentiation. The mean null values obtained using the Raup–Crick beta diversity approach are consistent with the predictions derived from the operation of deterministic processes, indicating that habitat filtering plays a significant role in the studied karstic systems. Conducting experiments on tolerance (drought or flooding) across tree life stages with numerous local tree species would be necessary to clarify whether an environmental filter, e.g., the joint absence of competition and seed dispersal limitation, contributes to explaining the local species assemblage [8]. In addition, complementary observations could be carried out to evaluate the habitat preference and physiological response of seedlings in flooded and non-flooded habitats [22].

Our results highlight flooding as an abiotic condition that restricts the establishment and development of species that may disperse from upland forests to lowland flooded flats, leading to a striking difference in their floristic composition. Gaining an understanding of the local-scale processes involved in tree species differentiation between the seasonally flooded forest and upland forest provides greater insight into the conservation value of these arboreal formations in southern YP [55,86]. Both forests maintain a high diversity of epiphytes (bromeliads and orchids) and endangered endemic tree species such as *C. cozumelensis*, *C. reflexiflora*, *G. sanctum*, and *L. xuul* [44,49–51]. However, the functioning of seasonally flooded forest and upland forest is at risk due to the decline in rainfall in southern YP, with predictions suggesting further decreases in the near future [46,47,87]. The decrease in precipitation in southern YP may lead to the homogenization of the floristic composition. On the one hand, this may be due to the possible reduction in the distribution area of species associated with flooding conditions (e.g., *H. campechianum*). On the other hand, it may be due to the gradual increase in the relative abundance and distribution area of the species that occur in upland forests that are better adapted to drier conditions. Another possible trend of change involves an increase in the survivorship and establishment in flooded forests of species typical of early successional stages, such as *Bursera simaruba*, whose populations have recently been reported to increase [44], even though it is not a tree species typical of seasonally flooded forests [41,86].

## 5. Conclusions

Waterlogging may act as an environmental filter favoring a differentiation of tree species composition in lowland floodplain and shallow hillslope karstic systems. This is reflected in the differentiation of tree species composition and the pattern of the Raup–Crick null model, which shows a limited influence of stochastic processes. These differences between the seasonally flooded forest and upland forest stands are remarkable considering the short distances separating them. Dispersal limitation (spatial distance) was found to be a minor driver in the assemblage of tree communities, although it did show a significant contribution. The flooding condition of the depressions may be at risk due to reduced rainfall in the YP region. Severe droughts can influence changes in floristic composition, slow down the development of forest structure, and, consequently, reduce the ability to mitigate the effects of climate change. We suggest conducting long-term in situ studies with a wide range of species in seasonally flooded forest stands. Most studies have focused on understanding changes in the dominance in functional traits with a perspective of the action of a double environmental stress, drought, and flood. Additionally, experiments related to seedling growth under anoxic conditions, aerenchyma formation in roots, and seed germination under flooded conditions can be conducted to better understand the floristic differences between flooded and non-flooded forests in karstic systems. Finally, understanding community assembly can support the design of ecological restoration plans



based on the selection of plants with traits related to environmental conditions, which can increase success and contribute to mitigating the effects of climate change.

**Supplementary Materials:** The following supporting information can be downloaded at: https://www.mdpi.com/article/10.3390/f15020250/s1; Table S1: Physical soil parameters (mean ± standard error) in seasonally flooded forest and upland forest. Values based on n = 9 samples for each forest type; Table S2: Description of topographic variables (mean ± standard error) of seasonally flooded forest and upland forest in the Calakmul region of the southern Yucatan Peninsula.

**Author Contributions:** Conceptualization, G.E.M.-A. and A.M.-R.; methodology, software, validation and formal analysis, G.E.M.-A., M.M.-T. and R.E.C.-S.; investigation and resources, G.E.M.-A. and A.M-.R.; data curation, G.E.M.-A. and R.E.C.-S.; writing—original draft preparation, G.E.M.-A., A.M.-R., M.G.-E. and M.M.-T.; writing—review and editing, G.E.M.-A., A.M.-R., M.G.-E. and M.M.-T.; visualization and supervision, G.E.M.-A., A.M.-R. and M.G.-E.; project administration and funding acquisition, G.E.M.-A. All authors have read and agreed to the published version of the manuscript.

**Funding:** This work was supported by the Comisión Nacional para el Conocimiento y Uso de la Biodiversidad (CONABIO) under Grant JF-128.

**Data Availability Statement:** Relevant data for this study can be obtained by contacting the authors in a reasonable manner.

**Acknowledgments:** We are grateful to the editors and reviewers for their valuable suggestions.

**Conflicts of Interest:** The authors declare no conflicts of interest.

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
