# Peer review of "Unraveling the Relative Contributions of Deterministic and Stochastic Processes in Shaping Species Community Assembly in a Floodplain and Shallow Hillslope System"

_forests, doi:10.3390/f15020250_

Round 1
Reviewer 1 Report
Comments and Suggestions for Authors
I was pleased to read this research article on the relative contributions of deterministic and stochastic processes in shaping species community assembly in a floodplain and shallow hillslope system. This study provides valuable insights into the ecological processes that drive species coexistence in different environmental conditions, shedding light on the mechanisms that shape diverse tree communities.
This manuscript reveals a significant floristic distinction between seasonally flooded and upland forests (1) and the influence of the topographic wetness index (TWI) on floristic differentiation (2). In addition, the dominance of deterministic processes over stochastic processes (3) and the potential phenotypic plasticity in response to flooding (4).
Overall, the four significant findings contribute to our understanding of how environmental factors, such as seasonal flooding and topographic variables, play a crucial role in shaping species coexistence and community assembly in contrasting abiotic conditions, providing valuable insights into the ecological processes driving tree communities in floodplain and shallow hillslope systems.
I wonder if authors could better describe the influence of higher spatial resolution DEMs for generating the TWI and also the use of other metrics as suggested by some authors [10.1016/j.geomorph.2023.108589; 10.3390/ijgi12080314; 10.1016/j.geomorph.2019.04.001; 10.1016/j.geomorph.2022.108316; 10.1016/j.foreco.2022.120639; 10.1016/j.foreco.2023.121244].
Other comments
Figure 1: Please add bar scale in the maps and north arrow; use colors and shadow relief for the maps;
L139: describe these DEMs with more detail and the data access for the general public (is it free for registered users? Nationals?)
L164-167: add that all sources are freely available;
L176: in R? detail;
L215-216: in R, right? Please add;
L219: data collection? Does it matter with the date of acquisition of the DEM? Detail;
Table 1: How do you set flood and upland?
In the discussion section, please add the possibility of applying more detailed spatial analysis tools to support the findings and impacts for conservations perspectives to justify publication in the esteemed Forests (MDPI) journal;
Conclusions contain a few pieces of information that were not profoundly discussed before in discussion. The discussion section could be split into subsections with more conservation, management, and sustainability implications.
Author Response
REVIEWER 1
I was pleased to read this research article on the relative contributions of deterministic and stochastic processes in shaping species community assembly in a floodplain and shallow hillslope system. This study provides valuable insights into the ecological processes that drive species coexistence in different environmental conditions, shedding light on the mechanisms that shape diverse tree communities.
This manuscript reveals a significant floristic distinction between seasonally flooded and upland forests (1) and the influence of the topographic wetness index (TWI) on floristic differentiation (2). In addition, the dominance of deterministic processes over stochastic processes (3) and the potential phenotypic plasticity in response to flooding (4).
Overall, the four significant findings contribute to our understanding of how environmental factors, such as seasonal flooding and topographic variables, play a crucial role in shaping species coexistence and community assembly in contrasting abiotic conditions, providing valuable insights into the ecological processes driving tree communities in floodplain and shallow hillslope systems.
I wonder if authors could better describe the influence of higher spatial resolution DEMs for generating the TWI and also the use of other metrics as suggested by some authors [10.1016/j.geomorph.2023.108589; 10.3390/ijgi12080314; 10.1016/j.geomorph.2019.04.001; 10.1016/j.geomorph.2022.108316; 10.1016/j.foreco.2022.120639; 10.1016/j.foreco.2023.121244].
R= We argue that HAND and TWI are simple hydrological-based models with robust premises that can reveal intrinsic relationships between relief parameters and water, providing new perspectives for the environmental assessment of small watersheds.
Other comments
Figure 1: Please add bar scale in the maps and north arrow; use colors and shadow relief for the maps;
R= Clarify; Figure 1 was modified adding colors, scale bar, north arrow and shadow relief map style.
L139: describe these DEMs with more detail and the data access for the general public (is it free for registered users? Nationals?)
R= Clarify, DEM was obtained from INEGI (accessed on 7 September 2023). DEM available for download free (https://www.inegi.org.mx/app/geo2/elevacionesmex/), no registration required.
L164-167: add that all sources are freely available;
R= Clarify, adding All modules used are open source and available free in…” in the manuscript.
L176: in R? detail;
R= Clarify, add R software citation. R software (version 3.6.2, R Core Team, http://www.R-project.org).
L215-216: in R, right? Please add;
R= Clarify, add R software citation. R software (version 3.6.2, R Core Team, http://www.R-project.org).
L219: data collection? Does it matter with the date of acquisition of the DEM? Detail;
R= Clarify, We update the web links to access the DEM and used software.
Table 1: How do you set flood and upland?
R= Clarify, we add Table S1 and S2 at the end of manuscript.
In the discussion section, please add the possibility of applying more detailed spatial analysis tools to support the findings and impacts for conservations perspectives to justify publication in the esteemed Forests (MDPI) journal;
R= We argue that HAND and TWI are simple hydrological-based models with robust premises that can reveal intrinsic relationships between relief parameters and water, providing new perspectives for the environmental assessment of small watersheds. The seasonally flooded forest is distributed in the southern portion of the Yucatan Peninsula. Its current state of conservation is due to the Calakmul reserves in its largest area of ​​distribution, the forest has been transformed for agricultural purposes (mainly sugar cane cultivation). Therefore, we consider that the scale of analysis is adequate for understanding ecological processes and in the future establishing ecological restoration programs.
Conclusions contain a few pieces of information that were not profoundly discussed before in discussion. The discussion section could be split into subsections with more conservation, management, and sustainability implications.
R= Clarify, we focus on arguing about the importance of ecological restoration in seasonal environments with the action of a double environmental stress (drought and flood).

Reviewer 2 Report
Comments and Suggestions for Authors
The manuscript entitled "Unraveling the Relative Contributions of Deterministic and Stochastic Processes in Shaping Species Community Assembly in a Floodplain and Shallow Hillslope System" evaluated the influence of topographic variables and spatial factors on the structure of tree communities in seasonally-flooded and upland forests in México.
The manuscript is well-written and structured. There are minor details to improve the current version:
Line 75: what is "a karstic landscape"? The authors introduce this concept for the first time in the text and they should clarify it for those readers unfamiliar with it.
The second hypothesis "...that topographic related variables to have a greater contribution than spatial factors in explaining the distribution of tree species" is not clear. Wouldn't spatial factors also be linked to topography? How can you separate both factors?
Please, clarify what statistical software to use for the analysis (I understand that it is R, but it is not explicit).
Replace "DBH>10cm" by "DBH≥10 cm"
Table 3: delete parenthesis. "Stems ≥10 cm" and "Stems < 10 cm"
Comments on the Quality of English LanguageNo comments.
Author Response
REVIEWER 2
The manuscript entitled "Unraveling the Relative Contributions of Deterministic and Stochastic Processes in Shaping Species Community Assembly in a Floodplain and Shallow Hillslope System" evaluated the influence of topographic variables and spatial factors on the structure of tree communities in seasonally-flooded and upland forests in México.
The manuscript is well-written and structured. There are minor details to improve the current version:
Line 75: what is "a karstic landscape"? The authors introduce this concept for the first time in the text and they should clarify it for those readers unfamiliar with it.
R= We add the text "mosaic formed by plains and hills whose origin is limestone or carbonate rock", to clarify the concept of karstic landscape.
The second hypothesis "...that topographic related variables to have a greater contribution than spatial factors in explaining the distribution of tree species" is not clear. Wouldn't spatial factors also be linked to topography? How can you separate both factors?
R= Clarify, We verify the correlation (r) between the topographic and spatial variables. resulting in a high correlation between elevation and PCNM1; which were not significant in the RDA (Tabla 3).
Please, clarify what statistical software to use for the analysis (I understand that it is R, but it is not explicit).
R= Clarity that the data sets were analyzed with the R program. Add R software citation. R software (version 3.6.2, R Core Team, http://www.R-project.org).
Replace "DBH>10cm" by "DBH≥10 cm"
R= We use the symbol ≥ instead of >; in all cases in the manuscript.
Table 3: delete parenthesis. "Stems ≥10 cm" and "Stems < 10 cm"
R= We delete the parenthesis from the Table 3.

Reviewer 3 Report
Comments and Suggestions for Authors
I have carefully read ms 'Unraveling the Relative Contributions of Deterministic and Stochastic Processes in Shaping Species Community Assembly in a Floodplain and Shallow Hillslope System' by Mendoza-Arroyo et al. The study examined tree communities in Calakmul, México, comparing lowland floodplain and hillslope karstic systems. 7,050 trees across 152 plots were studied using seven topographic variables. Analyses revealed distinct tree compositions in flooded versus upland forests, with the Topographic Wetness Index significantly influencing floristic differences. Deterministic processes predominantly shaped community assembly, highlighting the impact of seasonal flooding as an abiotic filter. Preserving both flooded and upland forests is crucial for tree diversity in the Yucatan Peninsula's karst, and the Topographic Wetness Index aids in identifying flood-prone areas in shallow depressions.
Abstract: line 23 what is RDA? please, explain.
The introduction introduces the reader to the proposed topic and is generally well written.
However, paragraph 75-86 belongs to the next section, Materials and Methods, it should be moved.
Materials and Methods: Study area and forest types subsection describes well both the area of study and forests in the area but I am not very comfortable with figure 1. Can you adjust figure 1a and b?
line 118, 136: Unfortunately both table S1 (and table S2) are not accessible at the time of review.
line 127: Where can I see the tree inventory?
Results: line 219: 139 tree species I presume. It is not very clear.
Discussions are generally well written although they seem rather overrated
Conclusions: this section does not meet the standards required by the journal. Therefore, please rework this important section.
Author Response
REVIEWER 3
I have carefully read ms 'Unraveling the Relative Contributions of Deterministic and Stochastic Processes in Shaping Species Community Assembly in a Floodplain and Shallow Hillslope System' by Mendoza-Arroyo et al. The study examined tree communities in Calakmul, México, comparing lowland floodplain and hillslope karstic systems. 7,050 trees across 152 plots were studied using seven topographic variables. Analyses revealed distinct tree compositions in flooded versus upland forests, with the Topographic Wetness Index significantly influencing floristic differences. Deterministic processes predominantly shaped community assembly, highlighting the impact of seasonal flooding as an abiotic filter. Preserving both flooded and upland forests is crucial for tree diversity in the Yucatan Peninsula's karst, and the Topographic Wetness Index aids in identifying flood-prone areas in shallow depressions.
Abstract: line 23 what is RDA? please, explain.
R= Clarify RDA is short for redundancy analysis
The introduction introduces the reader to the proposed topic and is generally well written.
However, paragraph 75-86 belongs to the next section, Materials and Methods, it should be moved.
R= Clarify, the paragraph is dedicated to describing the general context of the landscape in the Yucatan Peninsula. The conditions of the study area are described later and Tables S1 and S2 were added for a better description of the conditions at the location of each plot.
Materials and Methods: Study area and forest types subsection describes well both the area of study and forests in the area but I am not very comfortable with figure 1. Can you adjust figure 1a and b?
R= Clarify; Figure 1 was modified adding colors, scale bar, north arrow and shadow relief map style.
line 118, 136: Unfortunately both table S1 (and table S2) are not accessible at the time of review.
R= Add both Table S1 and S2 at the end of manuscript.
line 127: Where can I see the tree inventory?
R= Clarify. Data are data hosted in https://www.snib.mx/ . CONABIO - Sistema Nacional de Información sobre Biodiversidad de México
Results: line 219: 139 tree species I presume. It is not very clear.
R= Clarify that it must say 139 tree species. We add the word "tree".
Discussions are generally well written although they seem rather overrated
Conclusions: this section does not meet the standards required by the journal. Therefore, please rework this important section.
R= Clarify, we focus on arguing about the importance of ecological restoration in seasonal environments with the action of a double environmental stress (drought and flood).

Round 2
Reviewer 3 Report
Comments and Suggestions for Authors
Add explanations of all abbreviations in the abstract. These are necessary for the reader.
I also note that the comments have not been taken into account. I mention that not all and the answers given I cannot agree with the authors. Please revise the entire manuscript.
Author Response
REVIEWER 3 – ROUND 2
Add explanations of all abbreviations in the abstract. These are necessary for the reader.
R= Clarify RDA is short for redundancy analysis. DBH is shot for diameter at breast height.
I also note that the comments have not been taken into account. I mention that not all and the answers given I cannot agree with the authors. Please revise the entire manuscript.
R= Clarify, we aggregated the manuscript, pointing out the changes made in the summary and conclusion. Additionally, we added the supplementary tables (S1, S2).
------------------- 0o0 ----------------------
REVIEWER 3 - ROUND 1
I have carefully read ms 'Unraveling the Relative Contributions of Deterministic and Stochastic Processes in Shaping Species Community Assembly in a Floodplain and Shallow Hillslope System' by Mendoza-Arroyo et al. The study examined tree communities in Calakmul, México, comparing lowland floodplain and hillslope karstic systems. 7,050 trees across 152 plots were studied using seven topographic variables. Analyses revealed distinct tree compositions in flooded versus upland forests, with the Topographic Wetness Index significantly influencing floristic differences. Deterministic processes predominantly shaped community assembly, highlighting the impact of seasonal flooding as an abiotic filter. Preserving both flooded and upland forests is crucial for tree diversity in the Yucatan Peninsula's karst, and the Topographic Wetness Index aids in identifying flood-prone areas in shallow depressions.
Abstract: line 23 what is RDA? please, explain.
R= Clarify RDA is short for redundancy analysis
The introduction introduces the reader to the proposed topic and is generally well written.
However, paragraph 75-86 belongs to the next section, Materials and Methods, it should be moved.
R= Clarify, the paragraph is dedicated to describing the general context of the landscape in the Yucatan Peninsula. The conditions of the study area are described later and Tables S1 and S2 were added for a better description of the conditions at the location of each plot.
Materials and Methods: Study area and forest types subsection describes well both the area of study and forests in the area but I am not very comfortable with figure 1. Can you adjust figure 1a and b?
R= Clarify; Figure 1 was modified adding colors, scale bar, north arrow and shadow relief map style.
line 118, 136: Unfortunately both table S1 (and table S2) are not accessible at the time of review.
R= Add both Table S1 and Table S2 at the end of manuscript.
line 127: Where can I see the tree inventory?
R= Clarify. Data are data hosted in https://www.snib.mx/
CONABIO - Sistema Nacional de Información sobre Biodiversidad de México
Results: line 219: 139 tree species I presume. It is not very clear.
R= Clarify that it must say 139 tree species. We add the word "tree".
Discussions are generally well written although they seem rather overrated
Conclusions: this section does not meet the standards required by the journal. Therefore, please rework this important section.
R= Clarify, we focus on arguing about the importance of ecological restoration in seasonal environments with the action of a double environmental stress (drought and flood).
